# T-Cell-Based Cellular Immunotherapy of Multiple Myeloma: Current Developments

**DOI:** 10.3390/cancers14174249

**Published:** 2022-08-31

**Authors:** Gary L. Simmons, Omar Castaneda Puglianini

**Affiliations:** 1Cellular Immunotherapies and Transplant Program, Massey Cancer Center, Virginia Commonwealth University, Richmond, VA 23298, USA; 2Department of Blood & Marrow Transplantation & Cellular Immunotherapy, H. Lee Moffitt Cancer Center & Research Institute, Tampa, FL 33612, USA

**Keywords:** BCMA, CAR T-cell, Multiple Myeloma, TCR, tumor associated antigen (TAAs)

## Abstract

**Simple Summary:**

Over the past two decades, there has been significant progress in the treatment of multiple myeloma. Starting with the approval of bortezomib and lenalidomide, followed by newer agents in the same classes, monoclonal antibodies, and most recently idecabtagene vicleucel and ciltacabtagene autoleucel, which are genetically engineered autologous T-cell-based therapies, our view of this disease has changed from incurable to controllable and potentially curable. In addition to multiple myeloma and B-cell lymphomas, T-cell-based therapies are also actively investigated in various types of hematological and non-hematological malignancies and are considered one of the most impactful evolutions in cancer therapeutics. This review aims to summarize existing data regarding the efficacy, toxicity, and management of unique adverse events in T-cell-based therapies that are both clinically available and under investigation. We will also address undergoing efforts to improve the survival outcomes of multiple myeloma patients through this treatment modality.

**Abstract:**

T-cell-based cellular therapy was first approved in lymphoid malignancies (B-cell acute lymphoblastic leukemia and large B-cell lymphoma) and expanding its investigation and application both in hematological and non-hematological malignancies. Two anti-BCMA (B cell maturation antigen) CAR (Chimeric Antigen Receptor) T-cell therapies have been recently approved for relapsed and refractory multiple myeloma with excellent efficacy even in the heavily pre-treated patient population. This new therapeutic approach significantly changes our practice; however, there is still room for further investigation to optimize antigen receptor engineering, cell harvest/selection, treatment sequence, etc. They are also associated with unique adverse events, especially CRS (cytokine release syndrome) and ICANS (immune effector cell-associated neurotoxicity syndrome), which are not seen with other anti-myeloma therapies and require expertise for management and prevention. Other T-cell based therapies such as TCR (T Cell Receptor) engineered T-cells and non-genetically engineered adoptive T-cell transfers (Vγ9 Vδ2 T-cells and Marrow infiltrating lymphocytes) are also actively studied and worth attention. They can potentially overcome therapeutic challenges after the failure of CAR T-cell therapy through different mechanisms of action. This review aims to provide readers clinical data of T-cell-based therapies for multiple myeloma, management of unique toxicities and ongoing investigation in both clinical and pre-clinical settings.

## 1. Introduction

Multiple myeloma (MM) is a malignancy of plasma cells with substantial genetic heterogeneity that is frequently preceded by premalignant monoclonal gammopathy of undetermined significance (MGUS) [1,2]. During the transition from MGUS to smoldering MM and symptomatic MM, the immune system is known to demonstrate progressive impairment [3,4,5]. Since the approval of bortezomib, the first proteasome inhibitor in 2003, many therapeutic agents through different mechanisms have been approved. Despite such therapeutic evolution over the past 20 years, this disease is still considered incurable with a low 5-year overall survival rate [6] and disease control after multiple treatment failure is an unmet need.

Cell based therapies beyond conventional allogeneic/autologous stem cell transplantation is an emerging field and is currently clinically applied mainly in hematological malignancies [7,8]. A BCMA (B Cell Maturation Antigen) targeted CAR T-cell therapy, Idecabtagene vicleucel (ide-cel), is the first example of such approach in MM [9]. Recent studies have discovered that the complex immunosuppressive environment is associated with MM development in the bone marrow microenvironment [10,11,12] and this may suggest important therapeutic implications of cell-based approaches in MM. In this review, we discuss major concepts of novel T-cell based cellular anti-myeloma therapies both in clinical settings and under investigation. We categorize them into genetically modified strategies (such as various CAR T-cells and T-cell receptor engineered T-cells) and non-genetically modified strategies (such as marrow infiltrating lymphocytes, gamma/delta T-cells), which are summarized in Figure 1, and review their most up-to-date progresses in the field.

## 2. Genetically Engineered T Cells

### 2.1. Chimeric Antigen Receptor (CAR) T Cells

CARs are synthetic molecules that contain antigen-recognition and T-cell signaling domains [13]. Antigen-recognition domains are single-chain variable fragments (scFv) made from antibodies. Signaling domains include co-stimulatory domains derived from proteins such as CD28 and CD137 connected to an activation domain such as CD3ζ [13]. CAR constructs can recognize either a single antigen or multiple antigens on malignant cells by the scFv. CAR T-cell therapies have shown unprecedent activity even among the highly refractory/pre-treated MM patients as shown in both the CARTITUDE-1 and the KarMMa studies using ciltacabtagene autoleucel (cilta-cel) and idecabtagene vicleucel, respectively, with remarkable response rates of 73% to 97% in patients with a median of six or more prior lines of therapy [14,15] (Figure 2). Based on these encouraging results, both ide-cel and cilta-cel have received FDA approval. In MM, the lead CAR T-cell target is BCMA, which belongs to the TNF receptor superfamily that is expressed on B cells and is critical for plasma cell long-term survival [16]. Several additional CAR T-cell therapies are in clinical development and here we summarize the clinical experience of anti-BCMA CAR T-cell in multiple myeloma.

#### 2.1.1. NCI

The US National Cancer Institute (NCI) performed the first-in-human clinical trial using T-cells expressing an anti-BCMA CAR (11 D5–3-CD828 Z) in MM [18]. The CAR was a mouse single chain variable fragment (scFv) (11 D5–3), using a CD28 co-stimulatory domain, and was transduced with a lentiviral vector [19]. Patients expressing BCMA >50% by IHC were given fludarabine and cyclophosphamide for 3 days followed by a dose escalation of CAR T-cells/kg at 0.3 × 10^6^, 1 × 10^6^, 3 × 10^6^, and 9 × 10^6^ [19,20]. Responses were dose dependent, only 20% achieved a partial response at doses 0.3 to 3.0 × 10^6^ CAR T-cells/kg), but at higher doses such as 9 × 10^6^ CAR T-cells/kg, the overall response rate (ORR) was 81%, and MRD negativity 50% [20]. Toxicities were common in patients receiving CAR T-cells and appeared dose dependent—the highest cell dose experienced greater toxicity, grade 2 (44%), grade 3 (25%) and grade 4 (13%) cytokine release syndrome (CRS) [20]. Two patients had high grade 3/4 CRS, and both had a high plasma cell burden in bone marrow, therefore patients were required to have <30% bone marrow plasma cells before infusion [20]. At the highest dose level of 9 × 10^6^ CAR-BCMA T-cells, six (38%) required vasopressor support and one required intubation [20]. Peak CAR T-cell levels occurred between days 7 and 14 along with CRS but were not different between high grade and low-grade CRS [20]. At the time of cell infusion, the ratio of CD8:CD4 was 1:1, but post-infusion the cells were CD8+ with more cells expressing markers of exhaustion and senescence [19,20]. BCMA levels in the serum were elevated in patients with active MM and decreased with response to therapy suggesting a marker of disease surveillance [20].

#### 2.1.2. UPENN BCMA

This BCMA-targeted CAR used a lentivirus vector and was composed of a fully human scFv with a human 4–1 BB and CD3ζ intracellular signaling domain. In phase 1, single center study 3 cohorts were tested with varying CAR T-cell doses and conditioning chemotherapy [21]. The following cohorts were tested: cohort #1 with cell dose of 1 × 10^8^–5 × 10^8^ CAR T-cells without conditioning chemotherapy; cohort 2 with cell dose of 1 × 10^7^–5 × 10^7^ CAR T-cells using cyclophosphamide 1.5 g/m^2^; and lastly cohort 3 with cell dose of 1 × 10^8^–5 × 10^8^ CAR T-cells with cyclophosphamide 1.5 g/m^2^ [21]. CAR T-cell manufacturing took four weeks and were given to patients over three days in split dosing—10% of dose given on day 0, 30% on day 1, and 60% on day 2 [21]. Twenty-five heavily pre-treated patients with a median of seven prior lines of therapy and a majority (96%) with one high risk cytogenetic abnormality received CAR T-cell infusion [21]. A total of twenty-one patients received all three doses and four subjects received partial (40% of planned) because of early CRS. CRS occurred in (88%) of patients and eight patients (32%) that received higher doses (1 × 10^8^–5 × 10^8^ CAR T-BCMA cells) developed higher grade 3/4 CRS assessed according to the Penn grading scale [22]. Neurotoxicity was seen in eight of twenty-five patients (32%) while 12% had grade 3/4 encephalopathy, which included one patient with posterior reversible encephalopathy syndrome (PRES), severe obtundation, recurrent seizures, and mild cerebral edema on MRI that improved with high dose steroids and cyclophosphamide [21]. Of note, all three patients with severe neurotoxicity had a large tumor burden as well as received the highest dose of CAR T-cells [21]. There were dose limiting toxicities such as cardiomyopathy, spontaneous hemothorax, coagulopathy and one progressive MM with candidemia septic shock [21]. The ORR was 48% across all cohorts, and the dose with the highest efficacy was 1 × 10^8^–5 × 10^8^ CAR T-BCMA cells with an ORR 55% [21]. Responses were associated with peak expansion of CAR T-BCMA and both expansion/responses were associated with more severe CRS [21]. Peak expansion was between days 10 and 14, then CAR T-BCMA cell levels by qPCR declined, although the CAR was still detected in 20 subjects at 3 months and in 14/17 at 6 months after infusion [21]. Like the NCI CAR T-cell, the biomarker soluble BCMA (sBCMA) was utilized, and three long term responders maintained low serum sBCMA concentration, indicative that this is a useful tool for myeloma progression assessment.

#### 2.1.3. Idecabtagene Vicleucel (bb2121)

Idecabtagene Vicleucel is a second-generation autologous CAR T-cell produced by transduction with a lentiviral vector and encodes an anti-BCMA single-chain variable fragment, a CD137 (4–1 BB) costimulatory motif, and a CD3-zeta signaling domain [23]. In an early non-randomized, multi-center, phase 1 study of bb2121 in adult patients with relapsed/refractory multiple myeloma (RRMM) patients received standard lymphodepleting fludarabine/cyclophosphamide for three days followed by cell infusion at escalating doses: 50 × 10^6^, 150 × 10^6^, 450 × 10^6^, or 800 × 10^6^ CAR T-cells [24]. Patients had received a median number of previous antimyeloma regimens of 7 (range 3 to 23). The analysis of the first 33 patients demonstrated an ORR of 85% with 45% of the patients achieved complete response (CR) and a median PFS of 11.8 months [24]. As in other BCMA CAR T-cell studies, a dose-dependent relationship was observed and with doses <150 × 10^6^ CAR T-cells there were no VGPR or better responses [24]. Twenty-five patients (76%) had CRS, grade 1/2 was reported in 70% and grade 3 CRS was reported in 6% and no grade 4 or 5 CRS was observed [24]. Neurological toxicities of grade 1/2 were reported in 13 (93%) patients; 3% had grade 4 neurotoxicity, which occurred 11 days post-infusion and resolved in less than one month [24]. Updated data presented at ASH 2020 included 62 patients that received bb2121 and demonstrated deep and durable responses with median OS of 34.2 months and median PFS of 8.8 months [25].

The phase 2 study of idecabtagene vicleucel (KarMMa trial) enrolled 140 patients and 128 received ide-cel after standard lymphodepletion with fludarabine (30 mg/m^2^/day) and cyclophosphamide (300 mg/m^2^/day) × 3 days followed by target doses of 150 × 10^6^, 300 × 10^6^, or 450 × 10^6^ CAR T-cells [14]. This patient population was heavily pre-treated; 84% were triple refractory, 26% were penta-refractory, and 35% had high risk cytogenetics [14]. At a median follow-up of 13.3 months, the ORR was 73% and CR or better response was 33%. Of those patients with CR or sCR, 33 (79%) were also MRD-negative at a sensitivity level of 10^−5^ [14]. As with other CAR T-cells, the higher dose levels produced deeper response and peak expansion occurred at median of 11 days post-infusion [14]. CAR+ T-cells persisted out to 12 months in 36% of patients and undetectable sBCMA levels after infusion correlated with depth of response [14]. The median time to first response was 1.0 month and median time to complete response was 2.8 months with a median duration of response of 10.7 months in all patients and 19 months for those achieving CR or sCR at the 450 × 10^6^ dose, however it was only 4.5 months for those with PR [14]. In a subgroup analysis to determine predictors for CR, a multivariate analysis identified IgG heavy chain, high sBCMA, and elevated prothrombin time negatively correlated with CR/sCR and high vector copy number in drug product as a positive correlate of CR [26]. The estimated median PFS was 8.8 months overall, and 20.2 months in those achieving an sCR/CR. The estimated median OS was 19.4 months with an OS of 78% at 12 months. Updated data presented at ASCO 2021 with a median follow up of 15.4 months showed a median PFS of 8.8 months in all treated patients and the median OS was not reached [27].

Most toxicities occurred within two months of CAR T-cell infusion and consisted of cytopenia’s and CRS [14]. There were four bleeding events (cerebral, gastrointestinal, conjunctival and postprocedural) [14]. CRS was reported in 104 patients (84%), mostly grades 1–2, less than 5% experienced grades 3 and 4, and one patient had grade 5 CRS [14]. The median time to CRS onset was 1 day with a median duration of 5 days. Tocilizumab was administered to 67 patients (52%) and glucocorticoids were also used in 19 patients (15%) [14]. Neurotoxicity occurred in 23 patients (18%); 3% experienced grade 3 and no grade 4 or 5 neurotoxic events were observed [14]. The median time to neurotoxicity was 2 days, with a median duration of 3 days [14]. Forty-four patients died during the study period and most were attributed to myeloma progression [14]. Three patients died less than 8 weeks after infusion and causes of death for these individuals were pulmonary aspergillosis, gastro-intestinal bleed, and CRS [14]. There were four bleeding events in the brain, gastrointestinal, conjunctival, and post-procedural from the study [14].

Other studies of ide-cel in earlier disease settings are ongoing. KarMMa-3 (NCT03651128) is a multicenter, randomized phase 3 study that was opened to evaluate ide-cel vs. standard regimens in patients with RRMM with the primary endpoint progression free survival [28]. Eligible patients have received 2–4 prior lines of treatments including daratumumab, an immunomodulatory agent, and a proteasome inhibitor before enrollment [28]. Idecabtagene vicleucel dose level was 450 × 10^6^ CAR+ T-cells after standard lymphodepletion with fludarabine/cyclophosphamide × 3 days. Bridging therapy during manufacturing is allowed. KarMMa-4 (NCT04196491) is studying dose-limiting toxicity and safety in patients with newly diagnosed multiple myeloma (NDMM) and high-risk disease, as defined by the International Myeloma Working Group (IMWG) criteria [29,30]. Standard flu/cy conditioning followed by Idecabtagene vicleucel is given after four cycles of induction chemotherapy [30]. KarMMa-7 (NCT04855136) an open-label, multicenter, phase 1/2 study that evaluates combination of ide-cel with iberdomide (CC-220), an oral, novel cereblon E3 ligase modulator with direct antitumor activity and BMS-986405, a gamma secretase inhibitor that blocks shedding of surface BCMA to enhance tumoricidal activity of BCMA-directed CAR+ T-cells in RRMM [31]. The primary endpoints are safety and dose-limiting toxicity.

#### 2.1.4. Bb21217

A BCMA directed the CAR T-cell with identical CAR construct as idecabtagene vicleucel but adds the phosphoinositide-3-kinase inhibitor bb007 to enrich T-cells of a less-differentiated phenotype, thereby lessening the senescent T-cells. The idea is that the CAR T-cells will endure and persist longer, translating into a longer duration of response [32]. In a multi-center phase 1 dose escalation trial (CRB-402—NCT03274219) of heavily pre-treated myeloma patients, median of six therapies (3–17) and 68% triple refractory, seventy-two patients have received lymphodepletion with fludarabine (30 mg/m^2^/day) and cyclophosphamide (300 mg/m^2^/day) × 3 days, followed by one infusion of bb21217 as of February 2021 [32]. A total of 12 patients received 150 × 10^6^, 14 patients received 300 × 10^6^ and 46 patients received 450 × 10^6^ CAR+ T-cells with median follow up for entire cohorts of 9 months [32]. CRS occurred in 54/72 (75%) patients at a median time of onset of 2 days and the majority was grade 1/2 in 51 patients while one grade 3 and two deaths were also reported [32]. Neurotoxicity developed in 15% of patients at a median onset of 7 days, eight developed grade 1/2, two grade 3, and one grade 4 [32]. Responses occurred with ORR of 69%, ≥ CR of 28% and median duration of remission 17 months. CAR+ T-cells persisted in 30/37 (81%) patients at 6 months and 9/15 (60%) at 12 months after cell infusion [32]. Fifteen patients were evaluable for MRD with ≥ CR and (93%) were MRD negative at 10^−5^ by NGS [32]. In the peripheral blood 15 days post-bb21217 infusion, there were higher CD8 + CAR+ T-cells expressing CD27 and CD28 which correlated with significantly longer duration of remission also suggesting that less differentiated, more proliferative CAR+ T-cells at peak expansion are associated with better responses clinically [32].

#### 2.1.5. LCAR-B38 M

LCAR-B38 M is a 4–1 BB co-stimulatory domain transduced with lentivirus vector directed against two BCMA epitopes bestowing greater avidity binding to express anti-BCMA CAR [33]. In the phase 1 Legend trial, 74 patients were treated with LCAR-B38 M T-cells in different centers across China, but different lymphodepleting chemotherapy regimens and cell-infusion schedules led to report results from two separate cohorts: one of 17 patients and another of 57 patients [34,35].

Seventeen patients in this trial received varying conditioning and dosing schedules [34]. In cohort one, cyclophosphamide (250 mg/m^2^/day) and fludarabine (25 mg/m^2^/day) × 3 days followed by three days of at increasing dosing: d0 (20%), d3 (30%) and d6 (50%) [34]. In another cohort the patients received cyclophosphamide alone (300 mg/m^2^/day) × 3 days followed by a single infusion of CAR T-cells on day 0 [34]. The ORR was 88%, with 13 patients (76%) achieving sCR [34]. The median PFS in all patients was 12 months [34].

Fifty-seven patients in cohort two received LCAR-B38 M CAR after lymphodepletion with cyclophosphamide 300 mg/m^2^/day × 3 days [33]. The CAR+ T-cell dose (median, 0.5 × 10^6^ cells/kg) was split into 20%, 30%, and 50% of total dose given over 7 days [33]. The ORR was seen in 88%, in breakdown 42 patients (74%) achieved CR, two patients (4%) VGPR and six patients (11%) PR [35]. In the patients that reached CR, 93% (n = 39) were MRD negative by 8-color flow cytometry [35]. The median follow-up was 19 months and median duration of response was 22 months. In the MRD-negative CR patients, 91% were alive and had 27-month median duration of response [35]. BCMA expression was not clearly associated with response [33]. There were seventeen patient deaths during the study and follow-up cycle, eleven from myeloma and six from infection, suicide, gastrointestinal carcinoma, pulmonary embolism, and acute coronary syndrome, respectively [35].

CRS was assessed by modified Lee criteria [36] and occurred in 90% of patients at median time 9 days and majority grade 1 (47%)/grade 2 (35%), (7%) grade 3, and one patient with grade 5 [33]. One patient had CRS grade 2 which was resolving, but subsequently developed respiratory distress that mandated endotracheal intubation on day 22 and unfortunately died [33]. Neurotoxicity was observed in only one patient that developed grade 1 aphasia, agitation, and seizure-like activity.

#### 2.1.6. Ciltacabtagene Autoleucel

Cilta-cel (JNJ-68284528) contains two BCMA-targeting single-domain antibodies [37]. CARTITUDE-1 (NCT03548207) was both a phase 1 b (safety) and phase 2 (efficacy) trial investigating cilta-cel in RRMM patients that received ≥3 prior regimens (refractory to a PI, IMID, and had received an anti-CD38 antibody) [37]. Patients received standard cyclophosphamide 300 mg/m^2^ and fludarabine 30 mg/m^2^ × 3 days and one infusion of cilta-cel at a target dose of 0.75 × 10^6^ (range 0.5–1.0 × 10^6^) CAR+ viable T-cells/kg [37]. As of February 2021, 97 patients received cilta-cel (of these, 29 in phase 1 b; 68 in phase 2) and these patients were heavily pre-treated with 87.6% being triple-class refractory, and 42.3% with penta-refractory disease [17]. The ORR was 97.9% with sCR rate of 80.4%, the median time to first response was one month and the time to complete response or better was 2.6 months [17]. The median duration of response was 21.8 months and 91.8% achieved MRD negativity at the 10^−5^ threshold which persisted for ≥6 months in 44.3% of patients and ≥12 months in 18% [17]. CRS occurred in 94.8% of patients, mostly grade 1/2 with median onset 7 days (range 1–12) after infusion [17]. CRS grade 3/4 incidence was 4% and one patient died from grade 5 (hemophagocytic lymphohistiocytosis) [15]. CAR T-cell neurotoxicity was reported in 21% and 9% were grade 3/4 [15]. After cilta-cel infusion, twenty-one deaths occurred during the study period, ten deaths were attributed to disease progression, six to treatment, and five were due to adverse events unrelated to treatment based on the investigator assessment [15].

In CARTITUDE-2 study (NCT4133636), the primary endpoint is MRD negativity and it investigates cilta-cel in multiple myeloma in diverse disease settings and explores outpatient delivery. Cohort A examines cilta-cel in lenalidomide-refractory after one to three prior lines and had no prior exposure to BCMA-targeting agents [38]. Bridging therapy is allowed after apheresis and single infusion (target dose 0.75 × 10^6^ CAR+ viable T-cells/kg) given 5–7 days after start of lymphodepletion like CARTITUDE-1. At data cutoff of 15 April 2021; 20 patients underwent treatment and ORR was 95%. 85% of patients had ≥CR, and 95% had ≥VGPR with median time to initial response of 1.0 month. Of those eligible for MRD assessment, 92.3% were MRD-negative at 10^−5^ [38]. Toxicity profile is like CARTITUDE-1, hematologic adverse events significant for cytopenia’s. CRS occurred with median time to onset 7 days, while this was reported in 95% of patients only 10% of patients experienced grade 3/4 CRS [38]. CAR T-cell neurotoxicity developed in 20% and all was grade 1/2. One patient had grade 2 facial paralysis at 29 days which lasted 51 days and no other movement or neurocognitive events were reported [38]. At a median follow-up of 9.7 months, cilta-cel resulted in early and deep responses in patients who met criteria for CARTITUDE-2 cohort A [38].

In CARTITUDE-2, cohort B, the eligible population is RRMM patients who received one prior line of therapy and had early disease progression within 12 months after ASCT (autologous stem cell transplant) or within 12 months after initiation of anti-myeloma therapy and were naïve to CAR T-cell or anti-BCMA treatment [39]. A single cilta-cel infusion was given after conditioning which was identical to cohort A to 18 patients [39]. The ORR was 88.9% and 27.8% achieved ≥CR, 66.7% achieved ≥VGPR, the median time to first response <1 month and time to best response was 1.4 months [39]. Of the patients with ≥3 months post-infusion, 5 (38%) achieved ≥CR and in 9 patients who were MRD-evaluable, all were negative at 10^−5^ [39]. Hematologic adverse events were again cytopenias similar to CARTITUDE-1 and CRS occurred in 15 (83.3%) patients with only one grade 4 CRS [39]. The median time to onset was 8 days consistent with the heavily pre-treated patients in the CARTITUDE-1 study [39]. One patient with high burden disease, progression despite bridging, and CRS grade 4 had experienced movement and neurocognitive toxicity grade 3, 38 days post-infusion [39]. The patient had a VGPR and had some relief of movement disorder with immune-directed therapy [39]. In cohort B, one infusion led to early and deep responses in patients that had early relapse and or treatment failure [39].

Newly started and registered trials are investigating cilta-cel in earlier lines of therapy and outpatient settings. CARTITUDE-4 (NTC04181827) compares cilta-cel with traditional treatments, pomalidomide, bortezomib and dexamethasone (PVd) or daratumumab, pomalidomide and dexamethasone (DPd) in lenalidomide refractory patients. CARTITUDE-5 (NCT04923893) is evaluating bortezomib, lenalidomide and dexamethasone (VRd) followed by cilta-cel, compared to VRd followed by lenalidomide and dexamethasone (Rd) in NDMM whom autologous transplant is not planned. CARTITUDE-6 (NCT05257083), which will look at daratumumab, bortezomib, lenalidomide and dexamethasone (DVRd) followed by cilta-cel compared to daratumumab, bortezomib, lenalidomide and dexamethasone (DVRd) followed by ASCT in NDMM. Comparisons of two clinically approved agents, cilta-cel and ide-cel, are summarized in Figure 2.

#### 2.1.7. Orvacabtagene Autoleucel (Orva-Cel)

JCARH125 is a second generation anti-BCMA CAR T-cell construct containing a fully humanized scFv binder, 4-1 BB co-stimulatory and CD3 z activation domains [40]. The orva-cel manufacturing process resulted in a product with less differentiated CAR T-cell with predominance of memory-like (CCR7 + CD45 RA-) and naïve-like (CCR7 + CD45 RA+) phenotype that has a superior production of cytokines as well as superior proliferation and persistence capacity [41]. The phase 1/2 EVOLVE study (NCT03430011) enrolled heavily pre-treated patients with relapsed and/or refractory MM to receive orva-cel [40]. The median lines of prior therapies in this study were six and all patients were refractory to their last regimen.

Over 100 patients have been treated in the EVOLVE trial where JCARH125 was given as single infusion on day 1 at escalating doses of 50, 150, 300, 450 and 600 × 10^6^ CAR+ T-cells [42]. The objective response rate was 91%, with a CR/sCR rate of 39% [42]. Good safety profile with CRS and neurotoxicity grade ≥3 in 2% and 4% respectively [42]. In the EVOLVE study, baseline sBCMA levels correlated with tumor burden. Patients with higher baseline levels of sBCMA were more likely to experience CRS and neurotoxicity as well as other AEs. Interestingly, ORR was not influenced by sBCMA levels but those with high baseline sBCMA levels were less likely to uphold a response at month 6 [43]. In February 2021, Bristol Myers Squibb informed that the development of orva-cel will not be pursued to focus on the clinical development of ide-cel.

#### 2.1.8. Zevorcabtagene Autoleucel (Zevor-Cel)

CT053 is an autologous, second generation fully human anti-BCMA CAR T-cell product. The CAR construct consists of the fully human anti-BCMA scFv, a 4-1 BB co-stimulatory domain and a CD3-zeta signaling domain [44]. Twenty-four patients have been treated in phase 1 studies [45] with ORR, CR/sCR and mDOR of 87.5%, 79.2% and 21.8 months respectively. The median PFS was 18.8 months. Hematological toxicities were the most common grade ≥3 AEs. Grade 1–2 CRS was reported in 62.5% of patients (4 Gr-1 & 11 Gr-2) and resolved in a median of 6 days. Nine patients received tocilizumab at low dose (4–6 mg/kg). Three patients (12.5%) had neurotoxicity (2 Gr-1 & 1 Gr-3). One death was reported due to treatment-related serious AE (bone marrow failure, and neutropenic infection) and PD [45].

CT053 is being studied in two sister studies LUMMICAR-1 (NCT03975907) in China and LUMMICAR-2 (NTC03915184) in North America. Under LUMMICAR-1, fourteen patients have received CT053 at two different doses (three patients in 1.0 × 10^8^ group and eleven patients in 1.5 × 10^8^ group) with good tolerance and no dose-limiting toxicities observed [46]. Again, the most common grade ≥3 AEs were hematological with all patients experiencing grade ≥3 neutropenia. No grade 3 or higher CRS or neurotoxicity was observed. The ORR for LUMMICAR-1 phase 1 was 100%, with 11 sCR, 2 VGPR and 1 PR. All sCR patients were MRD negative at the 10^−5^ level of sensitivity. No immunogenicity was detected at data cutoff [46]. The 12-month PFS was 85.7%. With a median follow-up of 13.6 months, the mDOR and the median PFS were not reached [46].

In the LUMMICAR-2 study, fourteen subjects had received zevor-cel infusion in the phase 1 b portion of the study, including eight patients who received 1.5–1.8 × 10^8^ CAR+ T-cells, and six patients who received 2.5–3.0 × 10^8^ CAR+ T-cells. With a median of six prior lines of therapy, 93% of the patients were triple class refractory and 64% were pentarefractory. All patients received bridging therapy [47]. Similar to LUMMINAR-1, no grade 3 or higher CRS or neurotoxicity observed. At a median follow-up of 4.5 months a 100% ORR was observed with 2 sCR, 2 CRs, 1 VGPR and 5 PRs. MRD was evaluable only in 85% of the patients (n = 12) at the 10^−5^ sensitivity level and of those, 91% of them were MRD negative [47].

#### 2.1.9. ALLO-715

The use of allogeneic CAR T-cells may not only reduce waiting time but also may theoretically provide the advantage of healthier T-cells with more functional potency, but allogeneic CAR T-cells are technologically complex requiring additional gene editing to eliminate endogenous TCR expression in order to reduce risk of graft-versus-host disease (GVHD). Additionally, allogeneic CAR T-cells can be rapidly eliminated by the host immune system and the need to often receive additional immunosuppressive agents in addition to standard lymphodepletion may increase the risk of prolonged and persistent cytopenias that may further increase the risk of infections [48]. As in other hematological malignancies, the use of allogeneic CAR T-cells in MM is under clinical development. At the 2020 ASH annual meeting, the results from the first-in-human allogeneic anti-BCMA CAR T-cell therapy ALLO-715 (UNIVERSAL study—NCT04093596) demonstrated the feasibility, safety, and preliminary efficacy of this treatment [49]. ALLO-715 is generated by lentiviral transduction with a second-generation CAR construct with a fully human scFv with high affinity to BCMA and featuring a rituximab-driven off-switch. The TCR alpha constant gene is disrupted to reduce the risk of GVHD and the CD52 gene is disrupted with Transcription Activator-Like Effector Nucleases (Talen^®^) technology to permit the use of ALLO-647, an anti-CD52 mAb, for selective and prolonged host lymphodepletion [50]. In the UNIVERSAL study, 42 patients were treated with ALLO-715 at four dose levels (40, 160, 320, and 480 × 10^6^ CAR+ T-cells) following different lymphodepletion regimens with fludarabine, cyclophosphamide and ALLO-647 [51]. Patients were heavily pre-treated with a range of 3–11 prior lines of therapy and 42.9% were pentarefractory. In patients who received the two highest dose levels, the ORR was 61.5% and ≥VGPR was 38.5%. The mDOR in patients that received the two highest dose levels was 8.3 months. CRS occurred in 52.4% of patients, with only 1 Gr-3. One patient had neurotoxicity with concomitant Gr-2 CRS. There were two Gr-5 events, fungal pneumonia, and adenovirus hepatitis [51].

Other off-the-shelf CAR T-cells for MM in clinical development include ALLO-605 (NCT05000450), P-BCMA-ALLO1 (NCT04960579), and UCARTCS1 A (NCT04142619). In order to improve efficacy of CAR T-cell therapy in RRMM, different strategies are being investigated including the targeting of other MM antigens such as G-protein-coupled receptor, class C group 5 member D (GPRC5 D) [NCT05016778—POLARIS, NCT05219721, NCT05325801]; signaling lymphocyte activation molecule, family member 7 (SLAMF7) [NCT04499339—CARAMBA-1, NCT03958656] as well as simultaneously targeting two different MM antigens such as CD19 and BCMA (NCT04236011, NCT04182581, ChiCTR-OIC-17011272). Selected landmark trials of BCMA-targeted CAR T-cells are summarized in Table 1.

### 2.2. Associated CAR T Cells Toxicities

During the clinical application of CAR T-cell therapy, several unique and serious adverse events, including CRS and neurological toxicities as well as prolonged cytopenias, infections, coagulopathy and on-target off-tumor toxicities (e.g., hypogammaglobulinemia), have been reported.

CRS is the serious acute toxicity most frequently associated with CAR T-cell therapy. The total incidence is close to 90%, yet severe, grade ≥3 CRS is reported less than 10% for most anti-BCMA products [14,15,20,21,33,40]; however, there are some exceptions, including the BCMA-targeted CAR T-cell constructs from UPENN and NCI with grade ≥3 of 32% and 38% respectively, due to host characteristics, high tumor burden, different co-stimulatory domain, and CAR T-cell dose [20,21]. T-cell-engaging therapies can set off an en masse storm of cytokines such as IL-6, IL-10, and interferon (IFN)-Υ, overwhelming homeostatic mechanisms and leading to CRS [52]. The massive release of IFN-Υ mediates the activation of other immune cells, most importantly macrophages [53]. Macrophage stimulation then results in the excessive release of cytokines such as IL-6, TNF-α, and IL-10 which later cause fever, fatigue, vascular leakage, cardiomyopathy, hypotension, and coagulopathy [52,53,54]. Several grading systems were developed over the years for better characterization of toxicities such as the Lee criteria, PENN criteria, CARTOX criteria and most recently, the standardized American Society Transplant Cellular Therapy (ASTCT) criteria published in 2019 [22,36,55]. The management of CRS is now standardized based on the severity, mostly supportive care with antipyretics, IV hydration, vasopressor support, and supplemental oxygen. Based on the grading, the management of CRS often involves the use of tocilizumab, and corticosteroids [55,56].

Immune effector cell-associated neurotoxicity syndrome, ICANS, is another toxicity specifically associated with CAR T-cell therapy. Overall incidence of any grade ranges between 18–32% and severe grade ≥3 is reported in 3–12% [14,15,20,21,35]. The mechanisms of ICANS are still under investigation. Current data suggest that this is likely mediated by systemic inflammation and cytokine production that drives endothelial activation and blood-brain barrier (BBB) disruption leading to elevated cytokine levels in the cerebral spinal fluid (CSF) [57]. Presentations are variable with most common of which include encephalopathy, headache, delirium, tremor, expressive aphasia, and less commonly confusion, seizures, brain hemorrhage, and cerebral edema are reported [57,58]. Grade 2 or higher ICANS are treated with corticosteroids [55,56]. Following treatment with ciltacabtagene autoleucel in CARTITUDE-1, 5% of patients experienced neurotoxicities described as movement and neurocognitive treatment-emergent adverse events (MNTs) [59]. These MNTs were categorized into movement disorders, cognitive impairments and personality changes and had median onset of 17 days (range 3–94) after recovery of CRS and ICANS [59]. One of these cases of MNTs included a patient that developed worsening parkinsonism approximately 3 months following cilta-cel infusion. In this case, persistent presence of CAR T-cells in peripheral blood and cerebrospinal fluid as well as lymphocytic infiltration in basal ganglia were reported [60]. Notably, the investigators demonstrated expression of BCMA on neurons and astrocytes in the reported patient’s basal ganglia, implying that the observed event could be on-target toxicity [60]. The risk of MNTs increased with high burden disease and grade >2 CRS or any associated neurotoxicity and strategies to reduce the risk include amending protocols for more aggressive bridging therapy and treating CRS and neurotoxicity at early onset [59].

The most commonly reported off-target adverse event is cytopenias, which is reported in the vast majority of patients with less frequent, severe complications such as major bleeding including gastrointestinal, cerebral and hemothorax [14,21]. BCMA is selectively induced during plasma cell differentiation but absent in naïve and most memory B cells; however, targeting BCMA still can lead to secondary hypogammaglobulinemia since non-malignant plasma cells can still be affected by CAR T-cells [18,60,61]. The possibility of insertional oncogenesis as off-target toxicity is not excluded and needs long term monitoring of patients treated with CAR T-cell [62].

### 2.3. TCR (T Cell Receptor) Engineered T Cells

Cancer associated antigens are frequently self-antigens and as a result approaches to enhance or transfer autologous T-cells with high affinity TCRs to these antigens have failed due to the deletion of such T-cell populations through thymic selection [63]. Adaptive T-cell transfer with ex vivo expansion of genetically engineered T-cell receptors is one of the strategies to circumvent this phenomenon. In contrast to CAR T-cells, which are based on T-cell activation downstream of TCRs and contain their own intracellular signaling domains, TCR-engineered T-cells are MHC restricted and need other costimulatory molecules to be present for activation and proliferation. On one hand, this enables TCR-engineered T-cells to recognize both surface and intracellular antigens through MHC antigen presentation which is advantageous over CAR T-cells, but it is accompanied by several downsides including inefficacy upon MHC downregulation on tumor cells [64] and mispairing of engineered TCRs with autologous TCRs which can potentially lead to unexpected adverse events [65]. In terms of target antigen selection, cancer testis antigens such as NY-ESO-1 and LAGE-1 (expressed in approximately 30% and 50% of MM patient samples, respectively [66,67]) are preferred targets to avoid on-target adverse events and have been investigated in clinical trials [68,69,70]. Below we summarize the available clinical data on TCR-engineered T-cell therapies in multiple myeloma.

#### 2.3.1. NCT01352286 (SPEAR T Cells)

In this study, investigators developed specific peptide enhanced affinity receptor (SPEAR) T-cells whose TCRs optimally recognize a peptide shared by NY-ESO-1 and LAGE-1 antigen presented by the allele HLA-A*02:01. The study enrolled 25 patients and they received SPEAR T-cells post-ASCT setting, 4 days after high dose melphalan and 2 days after ASCT [68,70]. Fifteen out of twenty-five patients were male and nineteen out of twenty-five were Caucasian. Median age was 59 (ranges 45–72) with 10/25 patients received more than four lines of prior treatment. The amount of transduced cell was <1 × 10^9^ in three, 1–5 × 10^9^ in twenty-one and more than 5 × 10^9^ in one patient [70]. In terms of efficacy, ORR was 80% at day 42, 76% at day 100 and 44% at 1 year with median PFS 13.5 months (95% CI, 8.9–31.1 months), median OS 35.1 months (95% CI, 22.7 months not reached) and median DOR 12.2 months (95% CI, 7.6–29.8 months) [70]. All patients experienced some adverse events. Most commonly reported AEs were diarrhea (96%), decreased appetite (2%), nausea (92%), fatigue (88%) and thrombocytopenia (88%). They are also commonly observed after ASCT. Of the patients, 96% experienced grade 3 or greater AEs and 28% were reported to experience any serious AEs related to study intervention. CRS was not reported but all patients experienced one or more AEs that can be confused with potential CRS symptoms. Such AEs are known to potentially mask the diagnosis of the milder form of CRS post-ASCT [68]. The most common AEs potentially related to CRS in grade 3 or higher were maculopapular rash (12%) and hypoxia (8%). In terms of neurologic events, no AEs consistent with encephalopathy, seizure or inflammatory polyneuropathy related to intervention were reported [70]. Functional studies revealed 23/25 patients maintained quantifiable SPEAR T-cells in peripheral blood at Day 100 post-infusion and 10/25 patients at 1 year. Cytokine production was also maintained up to 1 year post-infusion when post-infusion peripheral blood samples were tested ex vivo. Correlation between persistence/functionality and efficacy was unclear due to small sample size [70].

#### 2.3.2. NCT03399448 (NYCE T Cells)

The product studied, NYCE (NY-ESO-1-transduced CRISPR 3 X edited cells), was unique as the cells were not only transduced with lentiviral vector of TCR targeting NY-ESO-1 but also endogenous TCR and PD-1 were knocked out through CRISPR-Cas9 gene editing [69]. This additional gene engineering addresses two potential downsides of TCR-engineered T-cell approach: (1) persistently produced endogenous TCR α and β chains can pair with therapeutic TCR α and β chains which may lead to reduced expression of target TCR as well as unexpected adverse events through targeting normal tissue [65,71,72]; and (2) exhaustion or dysfunction of genetically engineered T-cells through PD-1/PD-L1 pathway [73,74]. The primary outcome was the feasibility of this multiple gene editing strategy. Two MM patients with refractory disease (one patient with eight lines of prior therapy and the other six lines including ASCT) were treated with this product. Lymphodepleting chemotherapy consisted of cyclophosphamide and fludarabine on days −5 to −3 followed by engineered T-cells infusion (1 × 10^8^/kg) on day 0. In terms of clinical outcome, one patient had SD and the other did not respond. None of the patients, including another patient that received the same product for liposarcoma, experienced CRS or overt AEs attributed to the intervention [69]. Transduction efficiency of engineered TCR ranged from 1.4–2.4% in the final products but the average half-life of the product was 83.9 days which is longer than other NY-ESO-1 specific TCR-engineered products (~1 week) [68,69,75,76]. The cells with successful CRISPR-Cas9 edition of TCR α chain and PD1 sustained persistence in peripheral blood (5–10% of all mononuclear cells) but those with TCR β chain did not engraft well likely related to the lowest level of editing efficiency at this locus [69]. In vitro experiments compared NYCE T-cells with NY-ESO-1 TCR transduced cells without additional gene editing to knock out autologous TCR, and PD-1 showed higher cytotoxicity in NYCE products. Single cell RNA sequencing was performed in one liposarcoma patient in vivo sample and this analysis showed an increase in genes associated with central memory, which is in contrast to the results with another NY-ESO-1 product which displayed characteristics of T-cell exhaustion [69,77].

#### 2.3.3. Miscellaneous

NCT01892293 is a phase I/II study designed to test the TCR-engineered product which recognizes the same peptide as NCT01352286 in relapsed or refractory MM without concurrent ASCT [78]. Unfortunately, they enrolled six patients but only two received the actual treatment and the study was terminated by the sponsor decision without any published data to date. NCT02457650 is another TCR-engineered T-cells trial targeting NY-ESO-1, however current status of recruitment is unclear, and no results have been published yet.

## 3. Non-Genetically Modified Strategies

### 3.1. Multi Tumor Associated Antigen Targeted T Cells (TAAs)

Despite proven high initial response rates, relapse after CAR T-cells or TCR T-cells through antigen escape is a clinical challenge [24,68,79,80]. Targeting multi tumor associated antigens have potential to retain anti-tumor activity by targeting multiple different pathways critical for tumor growth and this approach has been tested in both solid and hematologic malignancies with good safety profile and most of them did not need lymphodepletion therapy in contrast to engineered T-cell therapies [81,82,83,84,85]. The investigators expanded autologous, non-genetically engineered T-cells that target myeloma expressed target TAAs (PRAME, SSX2, MAGEA4, Survivin and NY-ESO-1) and infused them to 21 patients with MM [83]. This is a phase I study with primary objective to determine safety of this approach. They enrolled total of 23 patients with successful manufacturing and T-cells were infused to 21 patients (1 patient withdrew consent and another rapidly progressed before infusion and decided to be in hospice). Nine patients received experimental treatment as adjuvant therapy while in remission from last line of treatment and twelve patients had relapsed or refractory disease when they started study treatment. Twelve adverse events potentially related to the infusions were observed and only two events (leukopenia and thrombocytopenia) seen in the same participant were grade 3 or higher. This patient had baseline leukopenia/thrombocytopenia, and these numbers recovered to baseline without transfusion or growth factor support. The study was conducted in outpatient setting and although three patients were briefly hospitalized, none of them were deemed related to the intervention. Of the nine patients who were in CR at the time of infusion, only two have relapsed at months 7 and 13 while others remained in CR at a median follow up of 27.5 months. Of the 12 who had active disease at the time of infusion, two patients were early post-treatment at the time of censoring and response was not assessed. In ten patients who had response assessment, median PFS was 22 months. Compared to CAR T-cells with grade 3 or higher toxicities in 90% or more patients, this is significantly safer and the PFS of 22 months in patients with median of 3.5 prior lines of therapy is significantly longer than historical outcomes [83]. Another multi TAAs against MM was recently presented [84]. This is a phase I/II study testing T-cells target different sets of TAAs (WT1, CD138, CS1 and NY-ESO-1) and no dose limiting toxicities including infusion reactions, CRS, ICANS and other events that prolong hospitalization post-infusion was observed [84]. These studies are actively recruiting patients and updated results on efficacy as well as randomized controlled trials with these agents are necessary to validate clinical application.

### 3.2. Marrow Infiltrating Lymphocytes (MILs)

Tumor-infiltrating lymphocytes (TILs) in a tumor microenvironment are prognostic of better survival outcomes in different types of cancers [86,87]. As a source of adoptive cell transfer, TILs have advantage over peripheral blood lymphocytes with higher tumor specificity and activated cytotoxic phenotype [88]. The therapeutic potential of TILs was seen in melanoma [89], nasopharyngeal carcinoma [90] and non-small cell lung cancer [91]. Despite this advantage, TILs as anti-tumor therapeutic approach carries several limitations; need of surgical resection of tumors, limited number of patients have easily harvestable TILs and time to take TIL expansion [92].

In MM, bone marrow corresponds to tumor microenvironment in solid cancers. Compared to peripheral blood lymphocytes, an in vitro study revealed that marrow infiltrating lymphocytes (MILs) obtained from MM patients have capacity of greater expansion, enhanced antitumor specificity and increased migration towards SDF-1, a chemokine present in high concentrations in BM suggestive of better homing to BM. Interestingly, this cytotoxic effect extended to both mature myeloma plasma cells and their precursor cells evidenced by a significant inhibition in clonogenic assay [93]. The first clinical trial evaluating the feasibility of MILs in multiple myeloma was conducted in immediately post-autologous stem cell transplant as part of consolidation. 25 patients, both newly diagnosed and relapsed MM were enrolled. These patients were eligible for autologous stem cell transplant but did not achieve a complete response prior to transplant. MILs were collected prior to transplant, expanded ex vivo using anti CD3/CD28 beads + IL-2 and then infused on Day + 3 of transplant. All patients enrolled were successful in cell expansion in 7-day processing time, which is significantly shorter than TILs. Patients who achieved 90% or greater reduction in disease burden after MILs had significantly better PFS compared to those who did not (25.1 months vs. 11.8 months) [92]. Based on this result, two randomized clinical trials evaluating the efficacy of MILs in MM were initiated and both completed recruitment. NCT 01858558 evaluated the efficacy of adding MILs to maintenance therapy in high-risk myeloma patients post-ASCT and NCT 01045460 randomized myeloma patients post-ASCT to either MILs alone vs MILs + the allogeneic granulocyte macrophage colony-stimulating factor-based myeloma cellular vaccine. Their results are eagerly awaited in order to better design this treatment strategy.

#### γδ. T Cells

Vγ9 Vδ2 T-cells based on TCR are characteristic innate effector cells that comprise 1–10% subset of all circulating T-cells [94,95]. They were identified in the 1980s and are unique in their capacity to infiltrate into tumor tissues as well as HLA independent activation to exert tumor cytotoxicity. The activation of γδ T-cells is primarily dependent on exposure to phosphoantigens (isopentenyl pyrophosphate and 4-hydroxy-3-methylbut-2-enyl pyrophosphate) [96,97] or aminophosphonates (including bisphosphonates) [98] and both in vivo and in vitro activations also require IL-2 [99]. Of note, γδ T-cells exert cytotoxicity both in innate effector functions and antibody-dependent cell-mediated fashion [100,101]. Infiltrating γδ T-cells has been shown to be associated with better survival outcomes for several cancers [102,103].

In the case of MM, several pre-clinical studies have shown antitumor efficacy of γδ T-cells [104,105]. Caution is needed that the MM bone marrow microenvironment is very immunosuppressive on γδ T-cells. Exhaustion of γδ T-cells is already found in the MGUS phase and continues while patients are in remission from systemic therapy [106]. An example of such an immunosuppressive tumor microenvironment is the failure of the PD-1 blockade in MM [107], suggesting the presence of other humoral and cellular mechanisms to maintain the immunosuppressive milieu [108]. Ex vivo expansion of γδ T-cells showed safety and some antitumor efficacy in multiple myeloma [109]. Further investigation and different approaches are needed to optimize this strategy.

## 4. Conclusions

The clinical success of anti BCMA CAR T-cell therapy in MM led to excitement about how to best include this effective therapeutic strategy in a treatment algorithm, as healthier autologous T-cells are also known to be important in CAR T-cell expansion and anti-tumor effect. There are several clinical trials ongoing to address this question as well as combining CAR T-cell therapy with other anti-myeloma drugs to hopefully achieve a cure for a large proportion of myeloma patients. Clinicians have, however, encountered issues related to CAR T-cell therapies such as cost, disease control until the CAR T-cell is available for patients, as well as relapse after CAR T-cell. While continuing efforts to optimize the production and treatment algorithm of CAR T-cell therapy, therapeutic agents not targeting BCMA, such as CAR/TCR-engineered T-cell therapies targeting other neoantigens and the enrichment of MILs with intrinsic anti-myeloma reactivity, are worth attention and further clinical development. After decades of calling MM an incurable disease, we are finally approaching a time where we will counsel patients that this is now a potentially curable condition.

## Figures and Tables

**Figure 1 cancers-14-04249-f001:**
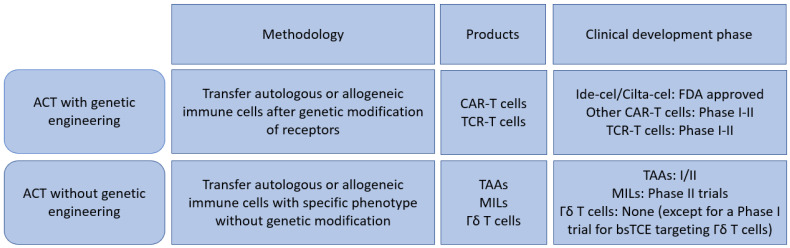
Summary of T-cell-based therapy for Multiple Myeloma. ACT: Adoptive cell transfer; CAR: Chimeric antigen receptor; TCR: T-cell receptor; TAAs: Tumor associated antigen targeted T-cells; MILs: Marrow infiltrating lymphocytes; bsTCE: bispecific T-cell engager.

**Figure 2 cancers-14-04249-f002:**
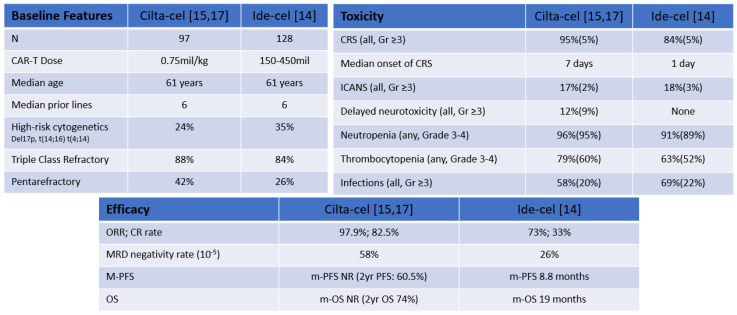
Comparable Features cilta-cel and ide-cel [14,15,17]. NR: not reached; m-PFS: median PFS; m-OS median OS.

**Table 1 cancers-14-04249-t001:** Summary of Selected Landmark Clinical Trials of BCMA-Targeted CAR T-cells. Mil: million; NR: not reached; scFv: single chain variable fragment; ORR: overall response rate; CR: complete remission; PFS: progression free survival; CRS: cytokine release syndrome; ICANS: immune effector cell-associated neurotoxicity syndrome; * results for “Dose Levels 3 and 4” as ORR and ≥VGPR.

Product	Phase	Specificity	scFv Binding Protein	Median Lines of Therapy	Dose	Response Rates (ORR/CR)	m-PFS	CRS (all/Gr ≥ 3)	ICANS (all/Gr ≥ 3)
Ide-cel [14] (*n* = 128)	II	Autologous	Murine	6	150–450 mil	73%/33%	8.8 months	84%/5%	18/3%
Cilta-cel [15,17] (*n* = 97)	Ib/II	Autologous	Camelid	6	0.75 mil/Kg	97.9%/82.5%	NR	95%/5%	17%/2%
bb21217 [32] (*n* = 72)	I	Autologous	Murine	6	150–450 mil	69%/28%	Not reported	75%/4%	15%/4%
Orva-cel [42] (*n* = 51)	I/II	Autologous	Human	6	300–600 mil	91%/39%	Not reported	2% (Gr ≥ 3)	4% (Gr ≥ 3)
Zevor-cel [47] (*n* = 20)	I	Autologous	Human	6	150–300 mil	94%/28%	Not reported	86%/0%	7%/0%
ALLO-715 (*n* = 42)	I	Allogeneic	Human	5	40–480 mil	61.5%/38.5%^*^	Not reported	52.4%/2%	2%/0%

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
