# Peer review of "T-Cell-Based Cellular Immunotherapy of Multiple Myeloma: Current Developments"

_cancers, 2022, doi:10.3390/cancers14174249_

Round 1

Reviewer 1 Report

The manuscript “T cell-based immunotherapy of Multiple Myeloma: Current Developments” by Gary L. Simmons and Omar C. Puglianini provides a comprehensive revision of t cell-based immunotherapies in MM, both engineering T cell and other T cell-based therapies. Th revision covers both academic and pharma therapies and explains each modality development, results and toxicities.

There are some details that should be improved/changed:

1 – In line 14 the sentence should be changed; the “and” in the beginning of line 14 is not understandable.

2 – in the abstract is said that CAR T cells was first approved in l B-cell lymphomas.  ALL should be also mentioned.

3 – In the Introduction, the authors should also mention efforts to use T cell based therapies in solid tumors.

4 – Also in Introduction some other recent publication in Cancers, as Raquel L. et al. could be added.

5 – Figure 2 should be citated earlier in the manuscript. My suggestion is after line 75 to allow comparison of ide-cel with cilta-cel when the trials are first mentioned.

6 – in line 79, I would add “both for academic and pharma”.

7 – The academic CAR T overview cells should cover the Barcelona project and not only UPenn and NCI.  Please add.

8 – The academic CAR T cells should be differentiated from the pharma ones. Information on which belongs to each company should be clear.

9 – The hurdles of the persistence of CAR T cells should be discussed.

10 – line 221, 222 – sentence is not clear.

11 – The comparison stated in line 254 should be removed as the two CAR T cells were not compared.

12 – Line 264, line 292, line 302 – misspellings/spacing should be corrected

13 – The sentence in the beginning of 2.2 needs to be concluded.

14 – The use of Tocilizumab and corticosteroids to deal with CRS is not rare. Please correct.

15 – Explain better the relation between PD-1 inhibition and gd t cells.

Author Response

Word document is uploaded

Reviewer 2 Report

This comprehensive and well-written review summarizes the state of the art of CAR-T cell treatment in multiple myeloma with special emphasis on BCMA as the target gene/protein. Furthermore, other T cell modifications, as well as non-genetically modified strategies, are explained nicely.

In my opinion, SLAMF7 that is investigated in the CARAMBA study should also be mentioned as a further therapeutic target. Furthermore, clinical studies for CD19 as a target are also ongoing and might be worth mentioning in the manuscript.
